# Neonatal apnea and hypopnea prediction in infants with Robin sequence with neural additive models for time series

**Julius Vetter**[1,2]*, **Kathleen Lim**[3,4], **Tjeerd M. H. Dijkstra**[5,6], **Peter A. Dargaville**[3,7], **Oliver Kohlbacher**[2,5,8], **Jakob H. Macke**[1,2,9], **Christian F. Poets**[4]

**1** Machine Learning in Science, University of Tübingen and Tübingen AI Center, Tübingen, Germany, **2** Department of Computer Science, University of Tübingen, Tübingen, Germany, **3** Menzies Institute for Medical Research, College of Health and Medicine, University of Tasmania, Hobart, Tasmania, Australia, **4** Department of Neonatology, University Hospital Tübingen, Tübingen, Germany, **5** Institute for Translational Bioinformatics, University Hospital Tübingen, Tübingen, Germany, **6** Department of Women's Health, University Hospital Tübingen, Tübingen, Germany, **7** Neonatal and Pediatric Intensive Care Unit, Department of Pediatrics, Royal Hobart Hospital, Hobart, Tasmania, Australia, **8** Institute for Bioinformatics and Medical Informatics, University of Tübingen, Tübingen, Germany, **9** Max Planck Institute for Intelligent Systems, Tübingen, Germany

* julius.vetter@uni-tuebingen.de

**Data Availability Statement:** The dataset analyzed in this study is available on Zenodo: https://zenodo.org/record/7711137. The code for all experiments

## Abstract

Neonatal apneas and hypopneas present a serious risk for healthy infant development. Treating these adverse events requires frequent manual stimulation by skilled personnel, which can lead to alarm fatigue. This study aims to develop and validate an interpretable model that can predict apneas and hypopneas. Automatically predicting these adverse events before they occur would enable the use of methods for automatic intervention. We propose a neural additive model to predict individual occurrences of neonatal apnea and hypopnea and apply it to a physiological dataset from infants with Robin sequence at risk of upper airway obstruction. The dataset will be made publicly available together with this study. Our proposed model allows the prediction of individual apneas and hypopneas, achieving an average AuROC of 0.80 when discriminating segments of polysomnography recordings starting 15 seconds before the onset of apneas and hypopneas from control segments. Its additive nature makes the model inherently interpretable, which allowed insights into how important a given signal modality is for prediction and which patterns in the signal are discriminative. For our problem of predicting apneas and hypopneas in infants with Robin sequence, prior irregularities in breathing-related modalities as well as decreases in $SpO_2$ levels were especially discriminative. Our prediction model presents a step towards an automatic prediction of neonatal apneas and hypopneas in infants at risk for upper airway obstruction. Together with the publicly released dataset, it has the potential to facilitate the development and application of methods for automatic intervention in clinical practice.

is available at https://github.com/mackelab/
neonatal_apnea_prediction.

**Funding:** This work was supported by the German
Research Foundation (DFG) through Germany's
Excellence Strategy, EXC number 2064/1–
390727645 and SFB1233 (PN 276693517); and
the German Federal Ministry of Education and
Research (BMBF): Tübingen AI Center, FKZ:
01IS18039A. The funders played no role in study
design, data collection, analysis and interpretation
of data, or the writing of this manuscript.

**Competing interests:** The authors have declared
that no competing interests exist.

## Author summary

Neonatal apneas (pauses in breathing) and hypopneas (shallow breathing) can severely impair infant development, especially in infants with conditions such as Robin sequence that increase the risk of upper airway obstruction. These breathing problems currently require frequent manual intervention by healthcare professionals, which can lead to alarm fatigue and challenges in providing consistent care. Here, we developed an interpretable machine learning model to predict these events in advance based on polysomnographie recordings. Our model achieved an average area under the receiver operating characteristic curve (AuROC) of 0.80 when discriminating breathing 15 seconds before apnea and hypopnea events from normal breathing. In addition, the interpretable additive structure of the model provided insight into which signal modalities of the polysomnographie recording were most predictive of adverse events. As such, this study provides a step toward automated solutions in neonatal care that could potentially improve safety and reduce the burden on healthcare providers in the future.

## Introduction

It would be of great clinical importance to be able to predict apnea and hypopnea events in neonates before they occur in order to perform preventive automated intervention. Automated intervention could be realized, for example, by an inflatable mattress through which stochastic vibrotactile stimulation can be applied [1].

Compared to the amount of work in automatic apnea *detection* [2–5], there has been considerably less work in automatic apnea and hypopnea *prediction* [6]. Prior work on prediction has mainly focused on associated events of bradycardia and used cardio-respiratory features together with different classifiers like hierarchical classification methods [7] and random forests [8]. Recent studies also used infant movement as an additional feature [6]. Another study more generally used deep neural networks to identify infants susceptible to apnea and hypopnea [9].

However, previous work did not address predicting individual events of apnea or hypopnea, but rather tried to predict whole episodes with repeated apneas and hypopneas [6]. Consequently, the prediction horizon for the cited approaches was on the order of several minutes [10, 11].

In this study, we were instead interested in predicting individual apnea and hypopnea events and thus worked on a time scale of seconds. Because of this new time scale, there were no readily available or traditionally used features to extract from the recorded signals. In recent years, deep neural networks have had considerable success in automating the feature extraction process. Many areas of science, including the prediction of adverse events in medical time series, have benefited from this progress [12–14]. However, a major drawback of classical deep neural network architectures is that they are a blackbox: Their complexity makes it difficult to understand why a particular prediction was made and which features of the signals contributed to it. Especially in the medical domain, where mistakes are costly, opening this blackbox and making its decisions more interpretable is crucial for applications. There are several methods to generate post-hoc explanations of black-box models, but their use in high-risk applications has recently been discouraged due to unreliable or misleading explanations [15, 16]. An alternative to post-hoc explanations are models that are interpretable by construction. Creating such models is often possible without sacrificing model performance [17].

### Our contribution

We built on recent work in neural additive models [18], which form their prediction by summing over the output of multiple neural networks. The additive nature allows users to inspect the magnitude of the additive contributions for each prediction to gauge the importance of the underlying feature or signal modality. This inherent interpretability makes neural additive models a good choice for clinical prediction tasks such as individual apnea or hypopnea prediction. Since our data came in the form of time series, we replaced classically used neural network architectures with those tailored to time series. We show that our model achieved a high level of performance, with an average area under the receiver operating characteristic curve (AuROC) of 0.80 for the task of apnea and hypopnea prediction for infants with Robin sequence. In addition, our model allowed us to assess the importance of different signal modalities for prediction and to visualize discriminative features within the signals.

## Materials and methods

### Data collection

We based our study on a physiological dataset from infants with Robin sequence. This group is well suited for automated prediction because of their homogeneous pathophysiology, that is, apneas and hypopneas are mainly caused by a narrow upper airway. Between May 2020 and April 2021, 19 infants with Robin sequence underwent whole night recordings in the Department of Neonatology at Tübingen University Hospital, using standard digital cardiorespiratory polysomnography (Remlogic, Natus Medical Incorporated, California, USA). Modalities recorded included nasal airflow obtained via a nasal pressure transducer and recorded at 200 Hz, as well as thoracic and abdominal respiratory efforts via respiratory inductance plethysmography recorded at 50 Hz. Furthermore, the heart was monitored via an electrocardiogram (EKG) and pulse plethysmogram (PPG) recorded at 200 and 100 Hz, respectively. The beat-to-beat heart rate was automatically derived from the EKG signal based on the RR-intervals. Finally, the $SpO_2$ and transcutaneous $PCO_2$ levels were recorded [19].

Five different types of adverse events were annotated in the recordings by a domain expert according to the criteria of the 2020 American Academy of Sleep Medicine (AASM) guidelines [19]. These were central apnea, obstructive apnea, and mixed apnea as well as central and obstructive hypopnea. Additionally, intermittent hypoxia events (desaturations) and body movements were annotated. Details about the annotation criteria can be found in S1 Appendix.

### Prediction setup

We framed the problem of apnea and hypopnea prediction as a binary time-series classification problem. For this purpose, all types of apneas and hypopneas were combined into a single type of adverse event. No attempt was made to distinguish between different types of apnea and hypopnea. Isolated intermittent hypoxia events were not considered adverse events. Parts of the signal annotated as either adverse event or movement were removed from the signal. The remaining signal was then divided into 30-second non-overlapping time windows. If a time window preceded an adverse event by an offset of 15 seconds, it was labeled as a target time window. This offset was well above the maximum duration of annotated apneas and hypopneas, ensuring that our prediction was not based on imprecise annotation time stamps. If a time window was at least three minutes away from both the start and end of an adverse event, it was labeled as a control time window (Fig 1). The choice of 30-second time windows was not arbitrary: Kelly and Shannon [20] define periodic breathing as "three or more episodes

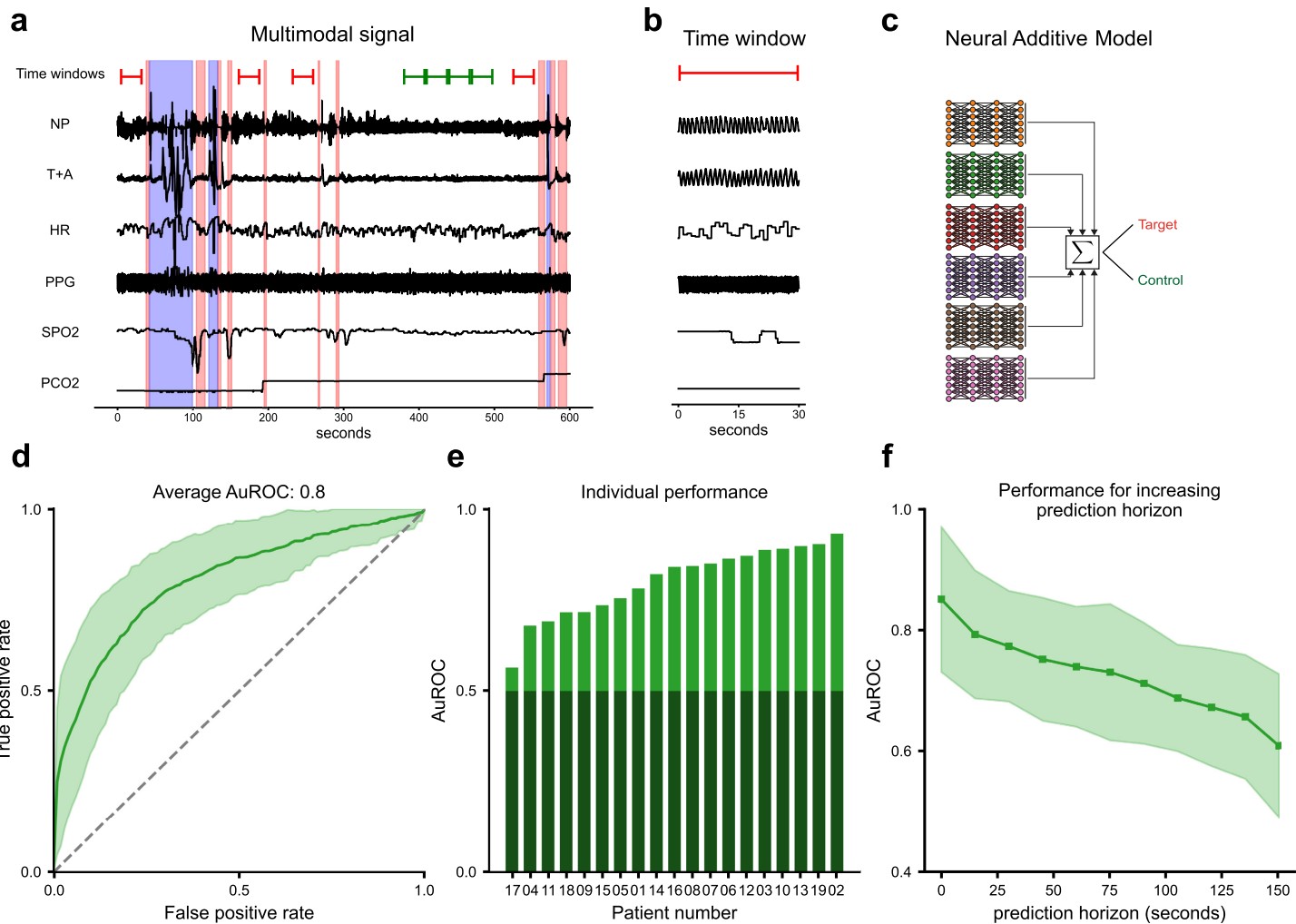

**Fig 1. Overview of the setup and core results. a)** 19 overnight respiratory polygraphy recordings of infants with Robin sequence were collected. Six signals were used for analysis: Nasal pressure (NP), the sum of thoracic and abdominal respiratory effort (T+A), heart rate (HR), photoplethysmogram (PPG), SpO₂, and PCO₂ levels. Adverse events (apneas and hypopneas, indicated as pink vertical bars) as well as infant movement (blue bars) were annotated manually. From these annotations, 30 second target time windows shortly preceding the adverse events (red), as well as control time windows (green) were extracted. **b)** Example of an extracted target time window. **c)** To discriminate between target and control time windows, a neural additive model (NAM) [18] was trained and tested with patient-based leave-one-out cross validation. **d)** Classification performance of the NAM: Pointwise average over the ROC curves of individual patients. Uncertainty corresponds to the pointwise standard deviation. **e)** AuROC for the individual patient-based test sets sorted in increasing order. Random performance is indicated in dark green. **f)** The average test performance as a function of an increasing prediction horizon. Uncertainty corresponds to the standard deviation over the performance of different patients.

of central apnea lasting at least 4 seconds, each separated by no more than 20 seconds of normal breathing". Therefore, this fixed length ensured a minimum distance between annotated adverse events to avoid the short inter-apnea episodes that occur during periodic breathing and are easily distinguished from normal breathing by their length alone [6, 21].

To evaluate the performance of the classifier, we performed patient-based leave-one-out cross-validation. That is, we used all extracted time windows of a single patient as a test set and performed the training and validation on the totality of all time windows extracted from the remaining patients. This procedure was repeated for each patient, and average performance scores and standard deviations are reported. For each iteration, we selected the hyperparameters using nested cross-validation [22]. This leave-one-out approach ensured that there was

no information leakage between the training and test sets, as the time windows on which the model was tested were extracted from a completely independent patient.

We focused on six signal modalities: Nasal pressure, the sum of thoracic and abdominal respiratory effort, heart rate (derived from EKG), PPG, $SpO_2$, and $PCO_2$ signals. Inter-breath intervals based on thoracic respiratory effort and heart rate have "classically" been used to predict apnea and hypopnea [6]. Since thoracic and abdominal respiratory effort are highly correlated signals, we treated them as a single signal by summing both signal traces pointwise. Creating the sum of these two modalities is a well-established practice in the analysis of apnea and hypopnea episodes [23, 24]. In addition, nasal pressure represents another interesting respiratory signal that may carry different information than the "classically" used thoracic and abdominal respiratory efforts. We also included the PPG signal and the two blood related parameters $SpO_2$ and $PCO_2$.

## Preprocessing

The three respiratory signals, nasal pressure, and thoracic and abdominal respiratory effort, sampled at 200 or 50 Hz, respectively, were downsampled to 5 Hz after application of an order 8 Chebyshev filter [25]. The same preprocessing was applied to the PPG signal. Both the $SpO_2$ and $PCO_2$ signals were downsampled from 2 to 1 Hz. The derived heart rate was left unchanged at 1 Hz. We then standardized all respiratory signals and the PPG signal on a time window basis. The summed signal of thoracic and abdominal respiratory effort was obtained by summing both standardized signals pointwise and standardizing the result again. Heart rate, $SpO_2$, and $PCO_2$ were range-normalized from 50 to 240 bpm, 60 to 100%, and 30 to 70 mmHg to a range of -1 to 1. No further preprocessing, data cleaning or missing value imputation was performed.

## Neural additive models for time series

Since there has been little prior work on possible features for individual apnea and hypopnea prediction [10], we performed automatic feature extraction with deep neural networks. To ensure interpretability, we used a neural additive model (NAM), which is a special case of a generalized additive model (GAM). GAMs combine expressiveness with built-in interpretability and have been used to analyze tabular data, especially in the medical domain [26–28]. For tabular data, GAMs are a linear combination of scalar, non-linear functions $f_i$:

$$\hat{p} = \sigma\left(\sum_i \alpha_i f_i(x_i) + \beta\right).$$

In binary classification, $\hat{p}$ denotes the classification probability and $\sigma(x) = (1 + e^x)^{-1}$ denotes the sigmoid function. GAMs have traditionally been fitted with splines or decision trees, but recent work has parameterized the functions $f_i$ with neural networks, giving rise to NAMs [18, 29]. Both GAMs and NAMs are inherently interpretable because it is possible to visualize each $f_i$ as a function of the corresponding feature. Moreover, given a specific data sample $x_i$, it is possible to analyze the individual importance of each feature by measuring the additive contribution $\alpha_i f_i(x_i)$ to the overall classification score.

Unlike GAMs, NAMs are not limited to tabular data. It is possible to choose $f_i$ to be any type of neural network architecture and thus to input whole time series instead of tabular values. When using more general networks to input entire time series $x_i$ instead of scalar features, it becomes difficult to visualize $f_i$. However, we can still measure the additive contribution of

each network $\alpha_i f_i(x_i)$ and thus still quantify the importance of a single time series modality to the overall classification.

To overcome the visualization difficulties, we parameterized the functions $f_i$ by fully convolutional networks (FCNs) [30, 31]. This allowed us to visualize discriminative features in the time series with activation maps [32].

## Architecture and training details

We implemented the presented NAM, as well as all downstream analyses in Python using the Pytorch [33] and the Scikit-learn [34] library. The individual subnetworks consisted of three convolutional layers. Kernel sizes were selected with nested cross-validation and ranged between 5 and 17. We used zero padding to preserve the time dimension of each input signal. Like the kernel size, the number of convolutional hidden channels was also selected with nested cross validation, which resulted in 20 hidden channels per layer. After each convolutional layer, we applied batch normalization [35] and rectified linear units (ReLUs) [36]. Before the global average pooling layer, network activations were linearly combined per time point to produce one single time series activation map for each signal modality.

The Adam optimizer [37, 38] was used to train the classifier with a standard binary cross entropy loss. To avoid training issues due to class imbalance, we undersampled the control time windows on a patient basis during training to balance control and target time windows. We trained the networks for 10 epochs with a learning rate of 0.0001 and a weight decay of 0.01. These choices resulted in optimal performance in every validation fold. The previously mentioned network hyperparameters were also stable across validation folds. This stability of hyperparameters across subjects was a consequence of our leave-one-out approach, where training sets differed only slightly between validation folds.

## Comparison to baseline models

As baseline models against which to compare our NAM, we trained a logistic regression and a multi layer perceptron (MLP) classifier, which perform the prediction based on a total of 24 engineered features. We engineered six features for the higher-frequency oscillatory modalities (i.e. nasal pressure, thoraric and abdominal respiratory effort, photoplethysmogram): Skewness and kurtosis of the signal in the time domain, as well as the centroid, spread, skewness and kurtosis of the power spectrum. For the three lower-frequency modalities (heart rate, $SpO_2$ and $PCO_2$ levels), we computed two features, that is, the mean and range (maximum minus minimum value) of the given time window.

Additionally, to quantify any trade-off in terms of model interpretability and prediction performance, we compared our NAM to a blackbox neural network model that works directly on the time series. We built this blackbox model by passing the high-dimensional features extracted by the individual subnetworks trough a MLP. Like the NAM, we trained this blackbox architecture end-to-end. Details on the feature engineering and baselines can be found in S1 Appendix.

To assess the significance of potential differences between different models, we always repeated our training procedure ten times to obtain average performances for each infant and computed Wilcoxon signed-rank ($p_{wil}$) tests over the $n = 19$ patients. Furthermore, we performed permutation tests [39] to check whether models perform significantly better than random. See S1 Appendix for more details on the statistical tests.

## Ethics

The ethics committee of the University Hospital Tübingen gave ethical approval for this work (application number: 352/2021BO2). The data was originally recorded for clinical purposes. The ethics committee of the University Hospital Tübingen gave permission to use and publish the anonymised data for research purposes without additional written informed consent from participants' parents.

## Results

### Descriptive statistics

We performed our analyses on a dataset of $n = 19$ infants with Robin sequence who were admitted to our Department of Neonatology between May 2020 and April 2021. Gestational age at birth was 39 (32–41) [median (range)] weeks, birth weight was 3,390 (1,320–4,380) g. At the time of respiratory polygraphy, infants were 17 (1–73) days old and weighed 3,392 (2,642–4,380) g. A total of 185 hours of respiratory polygraphy data were recorded, including 122 hours of total sleep time. Infants experienced 27 (3–112) obstructive, mixed and central apnea events per hour, and 28 (1–58) obstructive and mixed hypopnea events per hour, with events lasting for 3.8 (3.9–5.7) and 4.9 (2.8–7.3) seconds per event, respectively. After removing parts of the signals annotated as either adverse events or movements, the signals were divided into 30-second control and pre-adverse event time windows. This procedure resulted in 214 (75–606) time windows per infant to be used for classification. In all but one patient, there were more control time windows than target time windows. On average, 36% (4–66%) of windows corresponded to target events.

### Performance of neonatal adverse event prediction

Our neural additive model predicts neonatal apnea and hypopnea by performing a classification into pre-adverse event and control time windows. The average AuROC over all leave-one-out test sets was 0.80 with a standard deviation of 0.093 (Fig 1d). For some patients, we achieved classification performance above 0.9 AuROC (Fig 1e). We performed permutation tests [39] to assess statistical significance of test performance. For all but one patient ($p_{per} = 0.121$) the performance was significantly better than random ($p_{per} < 0.001$, see S3 Table for all permutation test results). Additionally, we computed true positive rates and precision scores at fixed false positive rates (FPR). For a fixed FPR of 20%, the NAM achieved an average true positive rate and precision of 70% (Table 1, see S4 Table for the full per-patient confusion matrices at 20% FPR).

We also investigated how much the performance altered when the prediction horizon between the target time window and the adverse event is increased or decreased from its default value of 15 seconds. As expected, as the prediction horizon increased, the classification performance decreased and reached near chance level after about 150 seconds. On the other hand, for no offset between annotated events and target time windows, the performance increased (Fig 1f).

**Table 1. True positive rate (TPR) and precision achieved by the NAM.** Both metrics were computed at different levels of false positive rate (FPR) aggregated over all 19 patients [median (25–75% quantile)].

|  | at 10% FPR | at 20% FPR | at 30% FPR |
|---|---|---|---|
| TPR (%) | 47 (33–66) | 70 (51–79) | 80 (60–87) |
| Precision (%) | 76 (60–85) | 70 (52–75) | 61 (45–68) |

Similarly, we investigated whether increasing the time window length would improve prediction performance. We tested this by doubling the time window length from 30 seconds to 60 seconds. We found that increasing the time window length did not significantly improve prediction performance (see S1 Appendix for details).

Our feature-based baselines, that is, the logistic regression and the multi-layer perceptron (MLP) classifier, also achieved a good average performance of 0.778 and 0.777, respectively. However, when comparing the interpretable NAM with the interpretable logistic regression, the NAM was significantly better ($p_{Wil} = 0.014$). Interestingly, the logistic regression, which linearly combines the 24 features was not worse than the MLP classifier, which is able to learn a complex, non-linear function of the same features. This is likely due to the fact that some of the information contained in the different modalities is redundant. We observed the same for the full blackbox neural network when compared to our NAM: The blackbox model was as good as our NAM, but did not achieve a significantly higher average performance ($p_{Wil} = 0.623$, S1 Table). In such cases, where many models achieve similar performance on a given problem, choosing inherently interpretable models is preferable [15–17]. See the Supporting information for additional information regarding the baseline model performances (S1 Fig and S1 Table).

## Neural additive model for signal modality importance

Next, we investigated how much predictive information the six physiological signal modalities carry for the classification. Our NAM classifier allowed us to investigate this by analyzing the additive contribution of each signal modality to the overall classification score. The higher the absolute value of an individual additive contribution is compared to the other contributions, the more influence it has on the overall classification. To assess the overall importance of the six signal modalities used for classification, we analyzed the additive contributions pooled across all patient test sets (Fig 2a). To analyze differences between patients, we calculated the standard deviations over the additive contributions of individual patients (Fig 2b). To

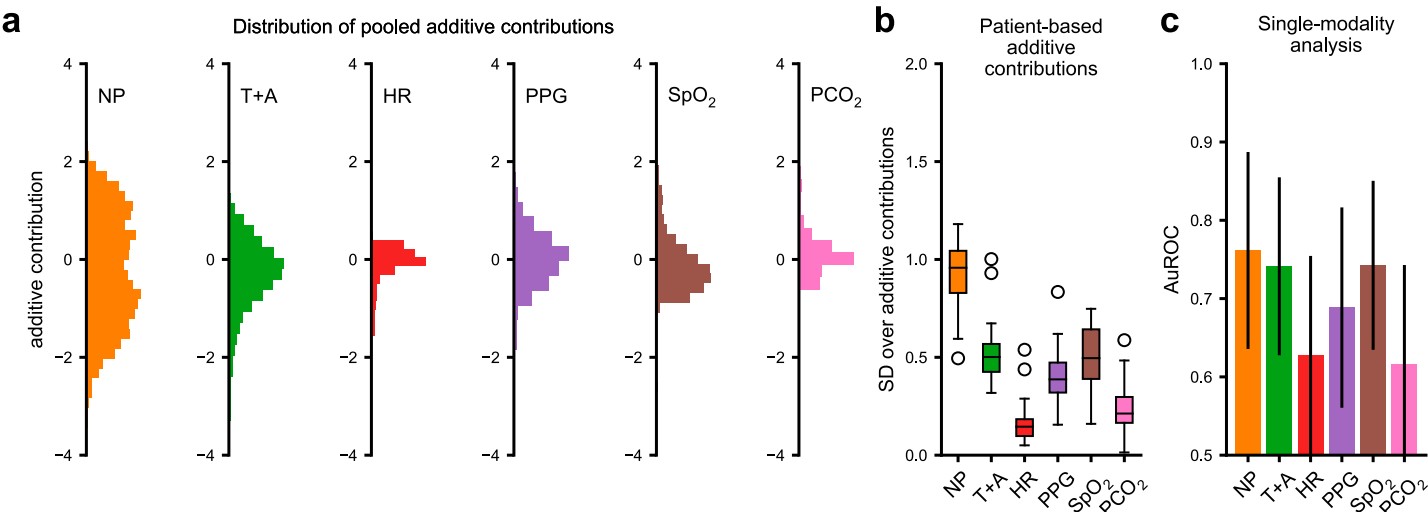

**Fig 2. Importance analysis of the signal modalities.** For the neural additive model (NAM), the importance of a signal modality for a given prediction can be measured by analyzing its associated additive contributions. **a)** Histograms of additive contributions pooled over all patient-based test sets (NP for nasal pressure, T+A for thoracic and abdominal respiratory effort, HR for heart rate, PPG for photoplethysmogram, and SpO$_2$ and PCO$_2$ levels). **b)** Box plots of the distribution over the standard deviation (SD) of additive contributions of all individual patients. **c)** Average AuROC scores of the individually trained single modality networks together with the standard deviation. All analyses show that the nasal pressure signal contains the most predictive information on average.

corroborate our analysis of the additive contributions, we also performed a single modality analysis, where the subnetwork of each signal modality is trained and evaluated independently.

The large standard deviations of both the pooled and patient-based additive contributions indicated that the nasal pressure signal contained the most predictive information (Fig 2a and 2b). The other respiratory signal, the sum of thoracic and abdominal respiratory effort, was comparatively less important for the prediction. This result was confirmed by the performance of the single modality networks: The nasal pressure network achieved the highest average AuROC, which was significantly better than the performance of the thoracic and abdominal effort network ($p_{wil}$ = 0.032, Fig 2c). The importance of both respiratory signals was consistent with the clinical perspective, as infants with Robin sequence mainly suffer from obstructive apnea and hypopnea. Furthermore, both the NAM and the single modality analysis showed that the other modalities heart rate, PPG, $PCO_2$ and especially $SpO_2$ carried predictive information in addition to the respiratory signals (S2 and S3 Tables). The performance of the single modality $SpO_2$ was comparable to that of the nasal pressure signal (no significant difference: $p_{wil}$ = 0.241). The power of $SpO_2$ was expected clinically: apneas and hypopneas often occur in clusters, and thus decreased $SpO_2$ values may indicate shortly preceding adverse events that increase the likelihood of another upcoming adverse event. A similar effect could be observed for heart rate, as apneas and hypopneas are often followed by bradycardia. In addition, the PPG signal is very sensitive to small movements, and thus can detect pre-apnea or pre-hypopnea arousal.

Although some of the signal modalities contained less information than others, the additive combination of their information increased the overall performance: The NAM achieved significantly better average AuROC than each of the single modality networks ($p_{wil}$ = 0.009, S1 Table).

Based on this modality importance analysis, we trained a NAM using only the three most predictive signal modalities, that is, the nasal pressure, the sum of thoracic and abdominal respiratory effort, and the $SpO_2$ level. The "reduced" NAM achieved an average AuROC of 0.802 that was not significantly different from the full NAM using all six modalities (S1 Table). As discussed in the comparison with the blackbox model, this result suggests some redundancy in the predictive information between modalities.

## Activation maps for local interpretability

While analysis of the additive contributions of the NAM and single modality performances provided insight into which of the signal modalities are relevant for model prediction, it did not provide information about which features of each signal modality were discriminative of an impending adverse event. For this purpose, we computed activation maps to visualize the discriminative features (Fig 3).

In our case, each time window classified by the NAM resulted in a total of six activation maps, one for each signal modality. Positive activations indicated that the corresponding segment of the signal was discriminative towards a target time window. In contrast, negative activations indicated that a segment of the signal was discriminative towards a control time window. Based on visual analysis, the classifier focused on irregularities in the respiratory signal for both nasal pressure and the sum of thoracic and abdominal respiratory efforts. In addition to these features, the classifier used arousals on the plethysmograph, decreased $SpO_2$ levels or $SpO_2$ desaturations, and heart rate variations to make a classification.

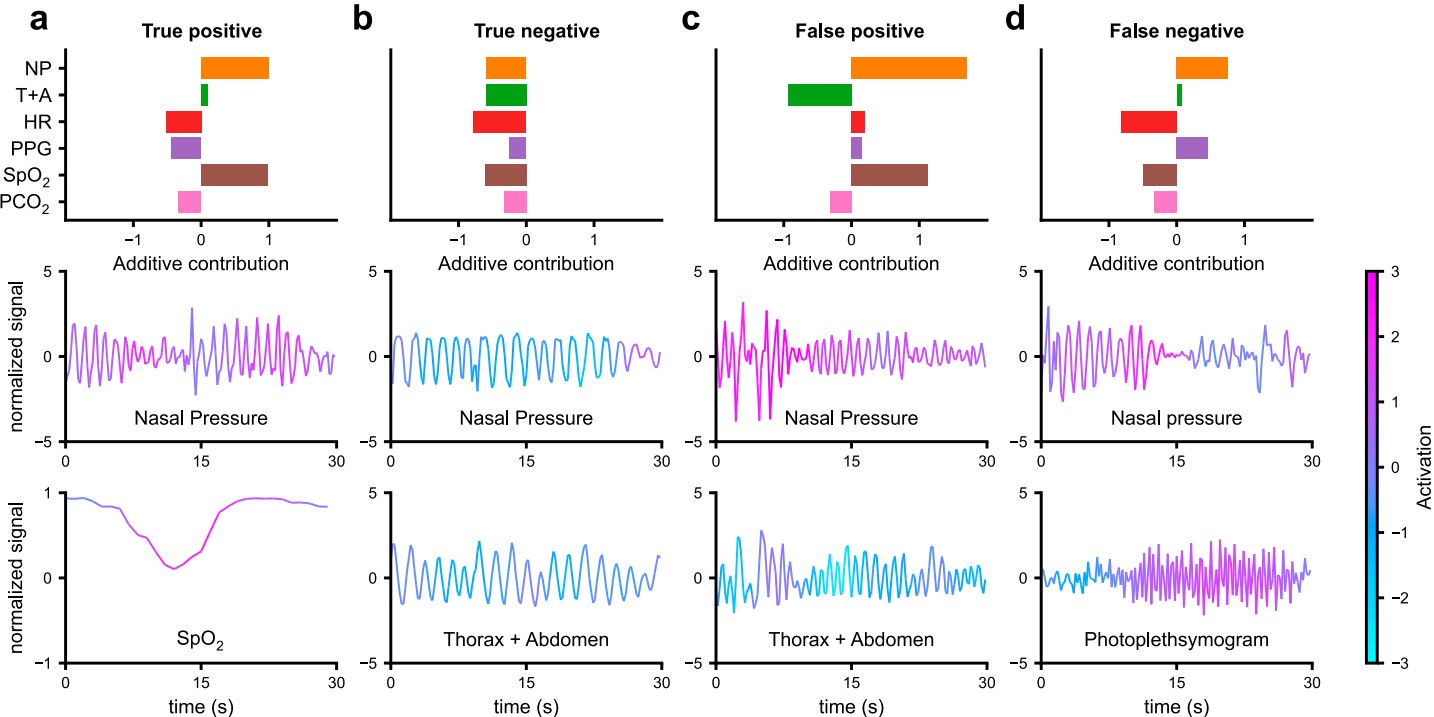

**Fig 3. Exemplary activation maps together with the associated additive contributions.** NP for nasal pressure, T+A for thoracic and abdominal respiratory effort, HR for heart rate, PPG for photoplethysmogram, and SpO₂ and PCO₂ levels. **a)** Correctly classified target (pre-adverse-event) time window. Light blue indicates negative activation and a classification towards no adverse event. Pink indicates positive activation and a classification towards an upcoming adverse event. The classifier detected different types of irregularities in the breathing signal as well as variations in heart rate and SpO₂ levels. **b)** Correctly classified control time window **c)** Incorrectly classified control time window (false positive). **d)** Incorrectly classified target time window (false negative).

## Discussion

We developed an interpretable time-series classifier and applied it to the problem of predicting neonatal apnea and hypopnea in infants with upper airway obstruction. In contrast to previous work, we focused on the prediction of single apnea and hypopnea events. Our neural additive model (NAM) was able to automatically extract relevant features from the multimodal polygraphy signal and achieved good performance in classifying individual time windows. Importantly, because of the inherent interpretability of the NAM, we were able to perform several downstream analyses to gain insight into the features used by the model.

First, we investigated the importance of each signal modality recorded with respiratory polygraphy. Instead of the classically used thoracic (and abdominal) respiratory effort [6], nasal pressure proved to be more informative in predicting individual events. This effect could be observed both in the additive contributions of the NAM and the performance of the single-modality networks. From a practical point of view, the predictive power of nasal pressure may allow clinicians to reduce the burden of respiratory polygraphy. If full respiratory polygraphy is not required clinically, monitoring nasal pressure alone may be sufficient to predict apneas and hypopneas. Alternatively, a NAM using only a subset of the modalities could be used. Second, we were able to visualize the learned features using activation maps. These visualizations allowed us to confirm their clinical significance.

The explanations provided by our model (additive contributions and activation maps) could play an important role in future clinical applications where nurses and NICU staff interact with automated apnea and hypopnea prediction systems. Unlike interpretable models

based on potentially complicated and unintuitive features (such as for example the spectral moments used in the logistic regression baseline), the explanations provided by the NAM add to the existing polysomnography recording to which practitioners are accustomed: The additive contributions show which of the recorded modalities were relevant to the prediction made, and the activation maps highlight the areas within the recording that were considered discriminative by the model. Based on these explanations, nurses and NICU staff have the capability to decide whether to trust or distrust the model's prediction. In the case of systematic false alarms, NICU staff could then intervene by recalibrating the model or, in extreme cases, turning it off altogether. Allowing for this type of distrust in a model is critical for use in real-world clinical settings [15].

The fact that the NAM works directly on minimally pre-processed polysomnography recordings also has computational advantages. In an automated system, the prediction itself could be realized almost instantaneously, leaving enough time to trigger potential interventions. In our study, we consider a prediction window of 15 seconds, which would be enough time to trigger interventions such as sensory stimulation [40, 41].

## Limitations

Although this study represents a first step towards the prediction of individual apnea and hypopnea events in neonates using machine learning, there are several obstacles that would need to be overcome in future work to allow the application of the presented model in a clinical setting. For some patients, the prediction performance was not sufficient to be of practical use. Especially for systems that perform automated interventions, high prediction accuracy will be necessary, as errors in a clinical setting may be costly: A warning system that produces many false negatives would not significantly reduce the stress an infant experiences from repeated apneas and hypopneas. Conversely, a warning system that produces many false positives would potentially place additional stress on the infant by prompting unnecessary interventions. Additional measures would also be needed to ensure the robustness of the model to unseen data, outliers, and malfunctioning recording devices to enable safe use.

Furthermore, in this study we were only concerned with performing "post-hoc" predictions of apneas and hypopneas. In a true interventional setting, the distribution of adverse events is subject to a distribution shift that may affect the performance of our current prediction model. We also excluded any periods of infant movement from analysis that would need to be dealt with in a real-world application. As such, this study represents a first step toward automated systems, but the practical validity of our model requires further study. Because our used dataset consisted of only 19 infants with Robin sequence who have a homogeneous pathophysiology, future studies need to apply the NAM to larger datasets and datasets collected in other settings with different patient demographics and more heterogeneous pathophysiology to provide external validation of our approach.

Finally, while the use of such automated machine learning systems in neonatal intensive care and health care in general is promising, the use of such systems raises challenging ethical considerations regarding (parental) consent, privacy, and accountability for potential harm.

## Conclusion

In this study, we developed a neural additive model that predicts individual events of neonatal apnea and hypopnea in infants with Robin sequence. Despite its limitations, our neural additive model represents a step towards automated prediction of neonatal apnea and hypopnea. If

reliable, such a model would have the potential to optimize the care of vulnerable neonates with upper airway obstruction.

## Supporting information

**S1 Fig. Comparison of neural additive model (NAM) to baselines models.**
(PDF)

**S1 Appendix. Details on the significance analysis, the baselines and feature engineering, as well as the description and definition of adverse events.**
(PDF)

**S1 Table. NAM vs. single modality networks vs. baselines.**
(PDF)

**S2 Table. Statistical test results for single modality network performances.**
(PDF)

**S3 Table. Permutation tests for both the NAM and all single modality networks.**
(PDF)

**S4 Table. Confusion matrices for individual patients.**
(PDF)

## Acknowledgments

We would like to thank the staff, parents and infants involved in this study, Ms. Gabriele Hilber-Moessner who provided additional support in obtaining the data, and Dr. Mirja Quante who provided valuable advice with annotating the data. We also thank the International Max Planck Research School for Intelligent Systems (IMPRS-IS) and the AI4Med-BW graduate program for supporting J.V.

## Author Contributions

**Conceptualization:** Julius Vetter, Jakob H. Macke, Christian F. Poets.

**Data curation:** Kathleen Lim, Christian F. Poets.

**Formal analysis:** Julius Vetter.

**Funding acquisition:** Oliver Kohlbacher, Jakob H. Macke, Christian F. Poets.

**Investigation:** Julius Vetter.

**Methodology:** Julius Vetter.

**Software:** Julius Vetter.

**Supervision:** Tjeerd M. H. Dijkstra, Peter A. Dargaville, Oliver Kohlbacher, Jakob H. Macke, Christian F. Poets.

**Validation:** Julius Vetter.

**Visualization:** Julius Vetter.

**Writing – original draft:** Julius Vetter.

**Writing – review & editing:** Julius Vetter, Kathleen Lim, Tjeerd M. H. Dijkstra, Peter A. Dargaville, Oliver Kohlbacher, Jakob H. Macke, Christian F. Poets.

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
