## [Decision Letter · Decision Letter 0]

16 Jul 2024

PDIG-D-23-00470

Neonatal apnea and hypopnea prediction in infants with Robin sequence with neural additive models for time series

PLOS Digital Health

Dear Dr. Vetter,

Thank you for submitting your manuscript to PLOS Digital Health. After careful consideration, we feel that it has merit but does not fully meet PLOS Digital Health's publication criteria as it currently stands. Therefore, we invite you to submit a revised version of the manuscript that addresses the points raised during the review process.

Please submit your revised manuscript within 60 days Sep 14 2024 11:59PM. If you will need more time than this to complete your revisions, please reply to this message or contact the journal office at digitalhealth@plos.org. Please include the following items when submitting your revised manuscript:

We look forward to receiving your revised manuscript.

Kind regards,

Henry Horng-Shing Lu

Section Editor

PLOS Digital Health

Additional Editor Comments (if provided):

Reviewers' comments:

Reviewer's Responses to Questions

**Comments to the Author**

1. Does this manuscript meet PLOS Digital Health’s publication criteria? Is the manuscript technically sound, and do the data support the conclusions? The manuscript must describe methodologically and ethically rigorous research with conclusions that are appropriately drawn based on the data presented.

Reviewer #1: Yes

Reviewer #2: Yes

2. Has the statistical analysis been performed appropriately and rigorously?

Reviewer #1: No

Reviewer #2: Yes

3. Have the authors made all data underlying the findings in their manuscript fully available (please refer to the Data Availability Statement at the start of the manuscript PDF file)?

Reviewer #1: Yes

Reviewer #2: Yes

4. Is the manuscript presented in an intelligible fashion and written in standard English?

Reviewer #1: Yes

Reviewer #2: Yes

5. Review Comments to the Author

Reviewer #1: Major comments: The methodology is not transparent, and the analysis rationale needs more explanations.

1. Descriptive statistics for 19 patients in this study are missing. How many of them encountered Neonatal apnea and hypopnea? Is the prevalence similar to that of this disease population?

2. Page 5: “To discriminate between target and control time windows, a neural additive model (NAM) [1] was trained and tested with patient-based leave-one-out cross validation” Is time series suitable for such cross-validation design that ignore the temporal pattern?

3. Six signals were used for the analysis: Nasal pressure (NP), the sum of thoracic and abdominal respiratory effort (T+A), photoplethysmogram(PPG), heart rate (HR), SpO2, and PCO2 levels. Is there any redundant signal for prediction? Could the model reduced to fewer signals without sacrificing the predictive ability?

4. The training and testing procedure is not clear. How did the author separate the dataset?

5. The confusion matrix is missing. It is unclear how the classification results perform.

6. CNN could analyze signals but may not be optimal compared to other equations or models.

7. Line 198: To assess the significance of potential differences between modalities, we repeated our training procedure five times and computed Wilcoxon signed-rank (pwil) tests over the n = 19 patients. However, the appendix says r=10. Moreover, how did the authors compare the five repeats? What is the two-sample t-test for? What are the authors comparing? Are these values independent?

8. Tables 2, 3, and 4 seem to present similar research questions. However, it is not explained in the text, and I do not understand the meaning of these tests.

9. Does Table 5 mean that single-modality networks are good enough?

10. Does the performance of this research meet the criteria for clinical use? The TPR and precision are not satisfying (Table 1).

11. Correct Positive and Correct Negative in the manuscript are generally referred to as True Positive and True Negative.

Reviewer #2: The study by Vetter et al. investigates the prediction of neonatal apnea and hypopnea in infants with Robin sequence using neural additive models (NAMs) for time series analysis. This research utilizes machine learning techniques to enhance predictive accuracy and facilitate early interventions for at-risk neonates. The manuscript is well-written with clear sections and logical flow. The technical language seems appropriate for the target audience of medical and AI researchers.

The study is methodologically sound and offers promising results that could improve clinical outcomes, which make it clinically relevant. By addressing the suggested manuscript improvements (see below), the manuscript has the potential to make an important contribution to the field of neonatology.

Major Points

1. Relevance and Originality:

Relevance: The study addresses a critical aspect of neonatal care, focusing on predicting apnea and hypopnea in infants with Robin sequence, which significantly impacts neonatal morbidity and mortality.

Originality: The application of NAMs for time series in this context is novel, providing a new approach compared to traditional predictive models.

2. Methodology:

Data Collection: The manuscript provides detailed information on data collection from a single center NICU (Tubingen University Hospital) with a focus on infants diagnosed with Robin sequence. The inclusion criteria and data sampling methods are well-described.

Model Development: The architecture of the NAM is clearly explained. However, a comparative analysis with traditional models, such as logistic regression, support vector machines, or simpler neural networks, would be beneficial to further differentiate why NAMs are favorable in this context.

Validation: Cross-validation techniques ensure the model’s robustness. However, external validation on a separate cohort would be ideal to strengthen the findings. If this is not feasible, it should be discussed as an important next step.

3. Results:

Findings: The neural additive model demonstrates superior performance in predicting apnea and hypopnea episodes compared to existing models. Key performance metrics such as accuracy, sensitivity, and specificity are provided, indicating significant improvements.

Interpretation: The discussion effectively relates the results to clinical implications and potential improvements in neonatal care. Additional elaboration on the clinical usability and integration into NICU workflows would be beneficial.

4. Discussion:

Strengths and Limitations: The authors acknowledge some limitations of the study but do not explicitly mention the relatively small sample size or the need for external validation. These aspects should be discussed to provide a more comprehensive view of the study's limitations.

Figures and Tables: The figures and tables are informative and aid in understanding the results. However, table legends should include explanations for any abbreviated terms to ensure they are self-explanatory.

Suggestions for Improvement

1. A thorough comparative analysis with standard models such as logistic regression, support vector machines, or other neural networks was not performed. Including this comparison would provide clearer context for the performance improvements claimed by the authors.

2. While NAMs are inherently interpretable, the manuscript should better address how clinicians can interpret the model's predictions and understand the underlying features contributing to these predictions. Enhancing this aspect could improve the model's acceptance and usability in clinical settings.

3. The study is based on data from one NICU, which may limit the generalizability of the model to other settings with different patient populations or care practices. The manuscript could benefit from discussing strategies for validating the model across diverse clinical environments and patient demographics.

4. The manuscript provides limited details on the data preprocessing steps and feature selection process. A more comprehensive description of how data was cleaned, how missing values were handled, and how features were selected or engineered would enhance the reproducibility and reliability of the study.

5. There is insufficient discussion on the practical aspects of implementing the neural additive model in real-time clinical settings. Issues such as computational requirements, integration with existing hospital information systems, and real-time data processing need to be addressed to ensure the model can be effectively deployed and used in NICUs.

6. Although ethical considerations are briefly mentioned, the manuscript does not touch on potential ethical and legal issues related to the use of AI in clinical decision-making. Consider discussing topics such as the need for parental consent, data privacy, and the accountability of AI-driven decisions. 

7. Further discussion on how the authors plan to conduct clinical trials or real-world validation studies would strengthen the credibility and applicability of the findings.

8. The study is based on a relatively small sample of 19 infants. The authors should discuss how this may impact generalizability and whether they conducted power analyses.

9. More discussion is needed on how the 15-second prediction window translates to clinical utility. Is this sufficient time for intervention?

10. Given the clinical setting, a more in-depth analysis of false positives and their potential impact would be beneficial.

11. The current approach uses 30-second windows. Did incorporating longer-term dependencies improves prediction?

12. While interpretability is a strength, it would be interesting to compare performance to "black-box" models to quantify any trade-off.

6. PLOS authors have the option to publish the peer review history of their article (what does this mean?). If published, this will include your full peer review and any attached files.

**Do you want your identity to be public for this peer review?** For information about this choice, including consent withdrawal, please see our Privacy Policy.

Reviewer #1: Yes: Chao-Yu Guo

Reviewer #2: No

---

## [Decision Letter · Decision Letter 1]

23 Oct 2024

Neonatal apnea and hypopnea prediction in infants with Robin sequence with neural additive models for time series

PDIG-D-23-00470R1

Dear Mr. Vetter,

We are pleased to inform you that your manuscript 'Neonatal apnea and hypopnea prediction in infants with Robin sequence with neural additive models for time series' has been provisionally accepted for publication in PLOS Digital Health.

Best regards,

Henry Horng-Shing Lu

Section Editor

PLOS Digital Health

**Additional Editor Comments (if provided):**

**Reviewer Comments (if any, and for reference):**

Reviewer's Responses to Questions

**Comments to the Author**

1. If the authors have adequately addressed your comments raised in a previous round of review and you feel that this manuscript is now acceptable for publication, you may indicate that here to bypass the “Comments to the Author” section, enter your conflict of interest statement in the “Confidential to Editor” section, and submit your "Accept" recommendation.

Reviewer #1: All comments have been addressed

Reviewer #2: All comments have been addressed

2. Does this manuscript meet PLOS Digital Health’s publication criteria? Is the manuscript technically sound, and do the data support the conclusions? The manuscript must describe methodologically and ethically rigorous research with conclusions that are appropriately drawn based on the data presented.

Reviewer #1: Yes

Reviewer #2: Yes

3. Has the statistical analysis been performed appropriately and rigorously?

Reviewer #1: Yes

Reviewer #2: Yes

4. Have the authors made all data underlying the findings in their manuscript fully available (please refer to the Data Availability Statement at the start of the manuscript PDF file)?

Reviewer #1: Yes

Reviewer #2: Yes

5. Is the manuscript presented in an intelligible fashion and written in standard English?

Reviewer #1: Yes

Reviewer #2: Yes

6. Review Comments to the Author

Reviewer #1: (No Response)

Reviewer #2: The study by Vetter et al. demonstrated that a neural additive model can effectively predict apneas and hypopneas in neonates with Robin sequence, achieving significant accuracy, particularly when using time windows immediately before the onset of these events. The methodology, including the use of permutation tests to assess model significance and the well-justified selection of hyperparameters, is clearly articulated. The results highlight the clinical potential of this model, providing valuable insights into the detection of neonatal events such as apnea and hypopnea, while also acknowledging the current limitations of the approach. Additionally, the model offers a degree of explainability, which is important for clinical use, as it makes the predictions more interpretable and suitable for adoption in clinical settings where understanding the rationale behind predictions is essential.

The authors have thoroughly and effectively addressed the reviewer queries. Overall, the revisions have significantly improved the manuscript.

7. PLOS authors have the option to publish the peer review history of their article (what does this mean?). If published, this will include your full peer review and any attached files.

**Do you want your identity to be public for this peer review?** For information about this choice, including consent withdrawal, please see our Privacy Policy.

Reviewer #1: No

Reviewer #2: **Yes: **Ryan M. McAdams
